# UNav: An Infrastructure-Independent Vision-Based Navigation System for People with Blindness and Low Vision

**DOI:** 10.3390/s22228894

**Published:** 2022-11-17

**Authors:** Anbang Yang, Mahya Beheshti, Todd E. Hudson, Rajesh Vedanthan, Wachara Riewpaiboon, Pattanasak Mongkolwat, Chen Feng, John-Ross Rizzo

**Affiliations:** 1Department of Mechanical and Aerospace Engineering, NYU Tandon School of Engineering, Brooklyn, NY 11201, USA; 2Department of Rehabilitation Medicine, NYU Grossman School of Medicine, New York, NY 10016, USA; 3Department of Population Health, NYU Grossman School of Medicine, New York, NY 10016, USA; 4Department of Academic Services, Ratchasuda College, Mahidol University, Nakhon Pathom 73170, Thailand; 5Faculty of Information and Communication Technology, Mahidol University, Nakhon Pathom 73170, Thailand; 6Department of Biomedical Engineering, NYU Tandon School of Engineering, Brooklyn, NY 11201, USA

**Keywords:** visual-based localization, VPR, weighted average, PnP, topometric map

## Abstract

Vision-based localization approaches now underpin newly emerging navigation pipelines for myriad use cases, from robotics to assistive technologies. Compared to sensor-based solutions, vision-based localization does not require pre-installed sensor infrastructure, which is costly, time-consuming, and/or often infeasible at scale. Herein, we propose a novel vision-based localization pipeline for a specific use case: navigation support for end users with blindness and low vision. Given a query image taken by an end user on a mobile application, the pipeline leverages a visual place recognition (VPR) algorithm to find similar images in a reference image database of the target space. The geolocations of these similar images are utilized in a downstream task that employs a weighted-average method to estimate the end user’s location. Another downstream task utilizes the perspective-n-point (PnP) algorithm to estimate the end user’s direction by exploiting the 2D–3D point correspondences between the query image and the 3D environment, as extracted from matched images in the database. Additionally, this system implements Dijkstra’s algorithm to calculate a shortest path based on a navigable map that includes the trip origin and destination. The topometric map used for localization and navigation is built using a customized graphical user interface that projects a 3D reconstructed sparse map, built from a sequence of images, to the corresponding a priori 2D floor plan. Sequential images used for map construction can be collected in a pre-mapping step or scavenged through public databases/citizen science. The end-to-end system can be installed on any internet-accessible device with a camera that hosts a custom mobile application. For evaluation purposes, mapping and localization were tested in a complex hospital environment. The evaluation results demonstrate that our system can achieve localization with an average error of less than 1 m without knowledge of the camera’s intrinsic parameters, such as focal length.

## 1. Introduction

According to the International Agency for the Prevention of Blindness, 295 million people are presently living with moderate-to-severe visual impairment and 43 million are living with blindness, a number projected to reach 61 million by 2050 [1]. Vision loss disproportionately affects multi-sensory perception when compared to other sensory impairments and has been shown to significantly decrease mobility performance and the ability to travel safely, comfortably, gracefully, and independently through the environment [2] Consequently, people with blindness and low vision (BLV) have difficulty traveling efficiently and finding destinations of interest [1].

Since the 1960s, numerous assistive technologies have emerged [3] to tackle travel difficulties. These technologies target context-awareness in the form of vision replacement, vision enhancement, and vision substitution [4]. Vision replacement aims to present environmental information directly to the visual cortex or optical nerve of the human brain. Vision enhancement techniques incorporate technologies such as augmented reality and artificial intelligence to restore the vision of the BLV. Vision replacement processes stimuli from other sensors and transmits them to a coupling system that converts them into nonvisual signals, often tactile, auditory, or a mix thereof. The focus of this paper is vision substitution, for which three subcategories of devices exist: Position Locator Devices (PLDs), Electronic Travel Aids (ETAs), and Electronic Orientation Aids (EOAs). PLDs determine the precise locations of the holder, and include technologies such as GPS, etc. ETAs are devices designed to detect near obstacles and to communicate the distance and the orientation of those obstacles relative to the end user. EOAs are devices that provide orientation and wayfinding information. Most of the commercial offerings in these categories have yet to gain significant market traction due to low accuracy, cost, and feasibility/implementation barriers, such as the need for physical sensor infrastructure.

This paper proposes a novel sensor-infrastructure-independent system for assistive navigation; the approach is cost-efficient and highly accurate, with an average error of less than 1 m. Our system is based on topometric maps computed by simultaneous-localization-and-mapping (SLAM) and structure-from-motion (SfM) algorithms. One distinct advantage of our system is a map-evolution feedback loop, in which query images from the target space are re-directed into a reference image database, accounting for dynamic changes in the target space and improving the density of the map data. Our system uses visual place recognition (VPR), weighted averaging, and perspective-n-point (PnP) algorithms for localization. More specifically, we adopt NetVLAD [5] for global descriptors and SuperPoint [6] for local descriptors to aid in the localization process. Once the localization is rendered, a suggested destination can be entered into a navigation pipeline and directions are generated. Navigation instructions are computed using Dijkstra’s algorithm and based on connecting the end user’s current location with a destination of interest. The system runs on a cloud server, which receives data as well as input commands and sends navigation instructions to the end user’s preferred mobile device over the internet. In cases of signal loss, our solution supports offline computation locally on the end-user device. This paper will discuss two types of end-user devices that we developed. One is an Android app based on Java language, and another is a backpack system with an Nvidia Jetson AGX Xavier and a bone-conduction headset.

The remainder of this paper is arranged as follows: a related-work section about sensor-based and vision-based navigation systems, a methods section that describes our system architecture and two end-user devices, an evaluation/results section, and, lastly, a discussion and conclusion section.

## 2. Related Work

Over the past three decades, many assistive technologies (AT) have been developed to help the BLV navigate independently and safely in unfamiliar environments [7,8]. These AT focused on navigation can be broadly divided into two groups: sensor-based and vision-based.

**Sensor-based devices**, which are dependent on pre-installed sensor input, are potential solutions for navigation pipelines [9]. However, all sensor-based technologies when translated at-scale, ensuring entire spaces are accessible, suffer from logistical issues, most importantly unrealistic economics. Devices that use Wi-Fi [10], Bluetooth [11], or ApriTag [12] require the pre-installation of beacons/modules and tedious calibration routines, driving up cost, maintenance, and inaccuracy. To tackle these issues, vision-based devices have been developed. Most use a smart mobile device equipped with a camera as a cost-efficient input sensor. In [13,14,15], the authors provide several summaries of current *state-of-art* vision-based localization solutions, which have two subcategories: retrieval-based localization and pose-based localization.

**Retrieval-based** localization, also known as image-based localization, uses a visual place recognition (VPR) algorithm to retrieve a set of reference images from a database that are visually similar to a query image taken by an end user, whose location can be estimated by extracting and averaging the geolocation of the retrieved reference images. The geolocations can be obtained either from a GPS or a 3D-reconstructed model. The VPR algorithm has two steps: feature aggregation and similarity search.

***Feature aggregation*** aims to represent an entire image as a low-dimensional vector assembled from the image’s feature points in order to accelerate searches when matching database images to a query image. BoVWs [16,17], VLAD [18], and DenseVLAD [19] are three traditional handcrafted feature aggregation algorithms that determine feature points by exploiting relations between each pixel of the image and its adjacent pixels. In 2016, NetVLAD proposed to use VLAD [18] in an end-to-end trainable deep neural network, and extracts feature points implicitly with the trained network. A series of evaluations has shown that NetVLAD outperforms the traditional handcrafted methods by a significant margin [5].

***Similarity search*** aims to find the similar reference images by isolating those whose low-dimensional vectors have minimal distances (e.g., Euclidean) to the query image’s vector through an exhaustive search. However, this search may be computationally expensive when the reference image database becomes large. To tackle this problem, the nearest-neighbor search method was proposed to reorganize the data’s store structure to speed up searching, as employed in a K-D tree [20], a hash table [21], or quantization frameworks [22,23], trading accuracy for rapidity.

**Pose-based** localization, unlike retrieval-based localization, calculates the more accurate six-DoF pose of the query image relative to the 3D space. There are three approaches in this class.

***The first approach directly regresses the pose from a single image using a deep neural network [24,25,26,27].*** The network implicitly represents a 3D reconstruction of the target space to retrieve the image’s pose. Evaluations have shown that, despite being efficient, the localization of this approach is often inaccurate [25].

***The second approach retrieves the query image pose by leveraging coarse prior information.*** This approach focuses primarily on refining the estimated coarse camera pose using pre-known geo-information obtained by GPS, Wifi, Bluetooth, a reconstructed 3D map, etc. In [28], the authors refine the camera pose by matching the extracted query image’s geometric features and building outlines with a GPS-obtained coarse prior pose. However, GPS signals are difficult to receive indoors, and WiFi, Bluetooth, etc. must be pre-installed and carefully calibrated, both of which create logistical challenges. In [29], the authors use a VPR algorithm to find a set of reference images similar to the query image and then refine the camera pose with a relative-pose computation algorithm. This algorithm, however, requires use of the camera’s intrinsic parameters, which contain the focal length information inherent to the specific camera being used, a step that is difficult to complete in an algorithm that must support multiple end-user devices.

***The third approach computes the camera pose by reprojecting 3D landmarks in a reconstructed map back to a 2D image and minimizing the discrepancies between the observed 2D points and their corresponding reprojections [30,31,32,33].*** Perspective-n-point (PnP) is the most frequently used algorithm to solve this reprojection, computing camera pose using a set of 2D–3D point correspondences between the query image and the reconstructed 3D map. However, it is time-consuming to search for 3D landmark correspondence in the reconstructed map for the 2D features in a query image. To improve computational efficiency, [34] introduced a coarse-to-fine strategy that first uses the VPR method to retrieve similar images of the query image and then uses the 3D landmark positions they stored to lessen the search range, enabling precise real-time localization in vast environments.

## 3. Method

In this section, our entire system architecture is introduced; then, we illustrate two types of user interface.

### 3.1. System Design and Architecture

Our system can be divided into three phases—mapping, localization, and navigation—as shown in Figure 1. Using a 360-degree field of view (FOV) camera (to improve the image database creation), a map-maker captures a video of a target space and extracts a sequence of equirectangular frames from this video. These sequential equirectangular images and the corresponding floor plan are fed into the mapping phase to generate a specialized ‘place’ map or so-called topometric map. This map is then used in the localization and navigation phases. Our system employs a VPR task to retrieve images similar to the query image $I_{q}$ taken by an end user, followed by two downstream tasks: weighted averaging and PnP to estimate the query image’s location and direction. Based on the retrieved location and direction, a shortest-path planning algorithm will safely guide the end user from an origin to a desired destination. We will explain these three phases in detail in the following subsections.

### 3.2. Mapping

The topometric map is generated in this phase, and plays a pivotal role in our entire system. It facilitates the delineation of boundaries around navigable spaces and the identification of destinations that may be of interest to end users. Furthermore, it contains a reconstructed 3D sparse map (or raw map) generated from multi-view RGB reference images of the target space and geolocations of these reference images, which are essential for estimating a camera’s location and direction from a query image. To reconstruct this sparse map, one could use the simultaneous-localization-and-mapping (SLAM) or structure-from-motion (SfM) algorithms. The former uses sequential images as input to generate the sparse map in real-time, while the latter uses unordered images and computes the sparse map offline.

OpenVSLAM [35] is a SLAM system based on Orb-slam2 [36] that supports multiple camera models, such as the equirectangular camera model, which has a 360-degree FOV and ensures sufficient overlap between adjacent images, which can enhance the robustness of the map reconstruction. It uses ORB features [37] to match two images, which works well when two images are relatively similar, but frequently fails when two images have large orientation or position differences. The SuperPoint network [6], on the other hand, can handle these differences robustly, resulting in a significantly more precise matching result. Colmap [38,39], one of the most popular SfM pipelines, supports SuperPoint features. However, it only supports the perspective camera model, which has less than $180^{\circ }$ FOV and therefore cannot guarantee sufficient overlap between adjacent images.

To ensure robustness of our system in both mapping and localization, we combine the advantages of OpenVSLAM and Colmap, as listed in Table 1. We construct a sparse map with OpenVSLAM and enhance it with Colmap by replacing its ORB feature with the SuperPoint feature. The input of our mapping module is a sequence of equirectangular images $I_{i}\in R^{3840\times 1920\times 3},\;\left(i=1,2,3,\cdots ,n\right)$ captured in the target space. Using these images, OpenVSLAM can accurately and robustly reconstruct a sparse map containing each equirectangular image’s 3D location $P_{i}$, direction $\alpha_{i}$, and a set of ORB features. We discard these ORB features and evenly slice $I_{i}$ into $m=\frac{360^{{}^{\circ }}}{\theta }$ perspective images $I_{i}^{t}\in R^{640\times 360\times 3},\left(t=1,2,\cdots ,m\right)$ with a width FOV of γ degree and a horizontal viewing direction of $\theta_{t}=t\times \theta$, where θ is the view direction intersection angle between two adjacent perspective images. These perspective images comprise a reference image database that is used in localization and navigation. For each reference image, we extract its SuperPoint features with local descriptor ${d^{l}}_{i}^{t}\in R^{1\times 256}$, compute its direction $\alpha_{i}^{t}=\alpha_{i}+\theta_{t}$, and send ${d^{l}}_{i}^{t},\alpha_{i}^{t}$, along with its location $P_{i}$, into Colmap to reconstruct the desired sparse map.

However, this sparse map is still defined in the 3D coordinate frame in OpenVSLAM (or Colmap), which lacks the necessary boundary information to ensure that end users navigate safely. To solve this problem, we project the sparse map onto a 2D floor plan’s coordinate frame using the transformation parameters between these two coordinate frames and define the relevant boundaries. To compute these transformation parameters, we need to find a set of 2D–3D point correspondences, which can be manually selected from our graphical user interface (GUI), as shown in Figure 2. When opening this GUI, all equirectangular images captured in the target space are loaded and can be individually selected from the list in *Zone 1* for browsing in *Zone 2*. Then, the map-maker can click the ‘Select Floor Plan’ button in *Zone 3* to upload the target space’s floor plan, which will then be displayed in *Zone 4*. To facilitate the selection of 2D–3D point correspondences, the map-maker can double left-click in *Zone 2* (and *Zone 4)* to open a magnified view of the currently selected equirectangular image (and the floor plan). To record a manually identified correspondence (such as two red dots shown in Figure 2), the map-maker can first click the feature point on the image and then click its corresponding location on the 2D floor plan. Note that each feature point has a 3D coordinate $X_{i}=\left(x_{i},\;y_{i},z_{i}\right)$ in the OpenVSLAM (or Colmap); therefore, a 2D–3D correspondence is identified. The *y*-axis of the OpenVSLAM (or Colmap) coordinate frame in our system is perpendicular to the ground plane, and can therefore be neglected from the coordinate transformation; all $y_{i}$ coordinates are set to 1. Once the map-maker selects h≥3 correspondences, we can use Equation (Equation 1) to calculate the transformation matrix,
(1)$$T=xX^{T}{\left(XX^{T}\right)}^{-1},$$Here, $x:R^{2\times h}$ means a set of 2D floor-plan coordinates, $X:R^{3\times h}$ means the set of corresponding 3D sparse map coordinates, and the resulting transformation matrix $T:R^{2\times 3}$ can convert coordinates from the OpenVSLAM frame to the 2D floor plan frame. Finally, using *T*, the locations of all reference images and the 3D landmark points in the sparse map can be projected onto the floor plan and displayed in *Zone 4* as red and green dots, respectively.

### 3.3. Localization

The locations of the reference images and the 3D landmark points are crucial to our end-user localization process. In contrast to the method in [29] discussed previously, we refine the camera location of the query image $I_{q}$ by averaging the locations of its top *K* similar reference images obtained via the VPR task. Similar to [34], we limit the searching range of the 2D–3D correspondences to only these similar reference images to speed up the computation; then, we use the PnP algorithm on the discovered correspondences to estimate the direction of $I_{q}$.

To retrieve images that are similar to a given query image from the reference image database, our system first uses NetVLAD to extract the global descriptors ${d^{g}}_{q}^{t},{d^{g}}_{i}^{t}:R^{1\times 32678}$ of $I_{q}$ and $I_{i}^{t}$; then, it calculates the Euclidean distance between them (Figure 3) using Equation (Equation 2)
(2)$$D_{qi}=\sqrt{\sum_{k=0}^{32767}{\left({d^{g}}_{qk}^{t}-{d^{g}}_{ik}^{t}\right)}^{2}},$$The lower the $D_{qi}$ is, the higher the similarity score between the reference image $I_{i}^{t}$ and the query image $I_{q}$. The reference images with the highest *K* scores (i.e., the lowest *K* Euclidean distances) are selected as similar or ‘candidate’ images $I_{j}^{t}\;\left(j=1,2,\cdots K\right)$. These candidate images are then utilized in two downstream tasks to estimate the end user’s location and direction.

The first downstream task uses a weighted averaging method to estimate the end user’s location by Equation (Equation 3)
(3)$$P=\sum_{j=1}^{K}\omega_{j}P_{j},$$

Here, *P* is the estimated location of the query image $I_{q}$. $P_{j}$ is the location of the candidate image $I_{j}^{t}$ on the floor plan and $\omega_{j}=\frac{m_{j}}{\sum_{k=1}^{K}m_{k}}$ is the weight applied on $P_{j}$, where $m_{j}$ (or $m_{k}$) is the number of matched SuperPoint local features between the query image $I_{q}$ and its candidate image $I_{j}^{t}$ (or $I_{k}^{t}$) using the SuperGlue network [40]. Note that $m_{j}$ will be set to 0 if it is not larger than 75. If $m_{j}$ of all candidate images are not larger than 75, the system will set *P* as the location of the candidate image with the largest $m_{j}$ that is larger than 30. If there is no $m_{j}$ larger than 30, the system will increase *K* and retry retrieval until it fails to estimate the camera’s location *P* when *K* exceeds a threshold.

The second downstream task efficiently estimates the camera’s direction using a coarse-to-fine strategy [34]. Specifically, the candidate image $I_{j}^{t}$ stores the 3D location of each of its 2D SuperPoint local features in the sparse map, after matching $m_{j}^{q}$ SuperPoint local features between $I_{q}$ and $I_{j}^{t}$ using the SuperGlue network. We are therefore able to obtain $\sum_{j=1}^{K}m_{j}$ 2D–3D point correspondences between $I_{q}$ and the sparse map, allowing us to efficiently calculate the camera’s direction using the PnP algorithm (Figure 4).

### 3.4. Navigation

After retrieving the current location and direction, the navigation module will guide the end user to the desired destination. A good navigation module should provide the end user with up-to-date boundary information for safe travel, as well as flexible and abundant destination options. We developed a GUI, as depicted in Figure 5, to define boundaries and destinations. It extracts all line segments from the floor plan image to represent potential boundaries and displays them on the topometric map in *Zone 1*, as in Figure 5. However, some boundaries might differ from the real world due to the quality of the floor plan or changes in the scene, requiring manual addition or deletion of boundaries in an interactive fashion. This GUI enables map-makers to maintain the map by removing or adding boundaries on the topometric map and redefining desired destinations quickly and efficiently. The left and the right areas shown in Figure 5 are magnified views of the floor plan displayed in *Zone 1*. The map-maker can remove boundaries in Figure 5 (left) when *Zone 1* is double left-clicked, or add boundaries or define destinations of interest in Figure 5 (right) when *Zone 1* is double right-clicked. Each green dot in *Zone 1* indicates a reference image in the database. To define a desired destination, the mapmaker must select any one of the reference images in the topometric map that are adjacent to an area of interest and assign it a destination name (Figure 5, right). Note that in our future work, we could utilize object/text detection methods to automatically detect each room’s number during the video capture and assign a destination to a reference image frame near that room.

Using the locations of the reference images as the potential destinations has accuracy and safety benefits. Because our localization method is based on VPR, which finds database images similar to a query image, our localization will become more and more accurate as the end user moves closer to the destination that is defined using the location of a database image, which increases the probability of successfully retrieving similar images to the query image. In addition, the reference images were captured by the map-maker, indicating that the area surrounding these reference images is navigable, thereby guaranteeing the safety of the BLV.

Due to the accuracy and safety benefits provided by the reference images, our system is designed to navigate the end user as closely as possible to the reference images. To accomplish this, it first determines if there exists an immediately navigable path between any pair of reference images by checking if there are boundaries between them. The paths calculated from all image pairs constitute a navigable graph, and Dijkstra’s algorithm is used to compute the shortest path between any pair of images based on this graph. This computation can be done off-line and the information can be quickly updated if the boundaries change. During the real-time navigation, the end user is required to select a desired destination from the destination list defined by the map-maker. Once the end user has been localized via a query image, the system will first direct the user to the closest reference image’s location, and then direct them along the shortest route to reach the destination.

### 3.5. User Interface

The end user can navigate using either of the two user interfaces we designed (Figure 6). One is for an Android application installed on the Android device, and the other is for a wearable device, which employs a discreet USB camera tethered to a micro-computer housed in a backpack.

#### 3.5.1. Android Application

The Android application contains a navigation bar, as shown in Figure 7. The end user needs to enter the server ID, port ID, and the number of seconds of automated camera acquisition (the end user can use the default settings without any changes), as well as select the current place, building, floor, and desired destination on their cell phone. Once the destination is selected, the phone’s camera will activate; the end user will need to hold the phone in landscape view and wait for the capturing time interval or tap the screen to capture a query image, which will then be sent automatically to a server to calculate current location and direction, after which a navigation prompt will be delivered. Our system supports another automated camera acquisition mode that intermittently takes the query image every predetermined number of seconds without requiring the end user to tap the screen. Note that the phone requires access to the camera, which can either be manually held (less preferred) or simply positioned in a lanyard at chest-level (more preferred). All touch-based operations in this application can be replaced with speech prompts to reduce operational difficulties for the BLV.

#### 3.5.2. Wearable Device

Cell phones are ubiquitous, but pose challenges when used for sustained periods, particularly when the camera feed is being used intermittently. In order to address the ergonomics of this problem and to improve image quality, we have developed a backpack [41,42,43] with an NVIDIA^®^ Jetson AGX Xavier connected to a battery, USB camera, and a binaural bone-conduction headset (Figure 6, right). The battery supplies power for the hardware; the USB camera is used to take query images. The end user can send vocal commands through the microphone in the headset and receive audio prompts from the server.

## 4. Evaluation

### 4.1. Overview

In this section, we present experimental evaluations of the developed system. There were six participants (four male and two female, with an average age of 32), including two end users with lived experience from blindness (one congenitally blind and another with degenerative retinal dystrophy), who participated in the evaluation process of the developed system (Figure 8).

### 4.2. Dataset

The evaluation was performed at an academic medical center in an ambulatory division within NYU Langone Health (*New York University Langone Ambulatory Care Center, New York, NY 10016, USA*).

A map-maker on our team used an Insta360 camera to collect equirectangular videos along a pre-designed ‘zigzag’ trajectory to ensure that the reference image database included maximal features from the target space. This trajectory included three loops in the target space. The first loop included the main hallway with all doors opened, the second loop included all hallways and the entrance into each room with doors opened, and the third loop included the whole space with all doors closed (meaning opened by the videographer during mapping). This process was designed through a trial-and-error process. We found the best camera height for pano-videos was approximately 6ft, given the distance between camera and ceiling in this particular space, attempting to capitalize on an aerial perspective while being mindful of proximity to the ceiling. By evaluation, it takes around 40 min to record an initial image database and around 15 min to build a topometric map for an area of about 264×900 feet. After extracting whole equirectangular frames (n=4258) from the video, we evenly sliced each of them into m=18 perspective images with θ = $20^{{}^{\circ }}$. Each perspective image has a size of 640×360 and an FOV width of $\gamma =75^{{}^{\circ }}$. These 18 images were filtered to avoid perceptual aliasing by counting the valid feature points extracted by the ORB detector; images were removed if the number of valid feature points fell below 100.

### 4.3. Localization Evaluation

Localization accuracy underpins navigation accuracy. Thus, we evaluated the localization accuracy of our system.

To test the system’s overall localization accuracy, we first selected 17 points on the floor plan as testing locations, which correspond to locations that are easily identified in the real world, such as the corner of structural columns and doorframes, and measured their pixel coordinates on the floor plan as ground truth locations. Each participant captured testing images at each testing location in the real environment with a ground truth direction obtained by a compass. The location/direction error was computed based on the Euclidean distance/absolute difference between the ground truth location/direction and the estimated location/direction. To draw a convincing conclusion, we averaged the error computed by all participants.

Since our system uses an image-retrieval method, there is a natural hypothesis that the denser the reference image database is, the more accurate the localization that can be achieved. To test this hypothesis, we designed two downsampling experiments based on the delineated dataset:***Frame downsampling*****.** We evenly downsampled the n=4258 equirectangular frames with a downsampling rate $\alpha \in \left[1,5,10,15,20,25,30,40,50\right]$ and sliced them into m=18 perspective images to form a reference image database;***Direction downsampling*****.** We maintained the original number of equirectangular frames equal to *n*. Then, after slicing each frame into m=18 perspective images and filtering into $\hat{m}$ valid slices, we evenly downsampled the slices with a downsampling rate $\beta \in \left[1,2,\cdots ,6\right]$ to form a reference image database.

## 5. Test Results

Due to our system’s single-thread processing design, the average localization time for each testing image was approximately 2 to 3 s when the number of image retrievals was set to 20. In the future, we will implement a parallel computing architecture to drastically shorten the localization time. This section is mainly focus on the localization accuracy testing. We calculate the location and direction errors at each of the 17 test locations based on the two experiments.

### 5.1. Localization Results

We visualize the localization error on a heat map (green indicates less error; red indicates more error) of our target space; each testing location is demarcated with an open circle, as shown in Figure 9. Here, we set both the α and β to 1, which means there were no downsampling operations on the original dataset. Under this configuration, the image database is the densest, but there is still a wide variation of location errors across 17 testing locations. The reason behind this phenomenon is that even though the reference image database (represented by the blue dots) is the densest, it remains challenging for map-makers to cover the entire floor plan when recording the reference video. Thus, when determining the location of a query image by applying the weighted average to the geolocations of its candidate reference images, the error will be large if the query image was captured in areas with insufficient reference images. To facilitate the analysis of the correlation between map density and estimated location precision, two tables based on the two evaluation settings are provided below. In these two tables, we examine the systematic decline in localization accuracy as a result of data downsampling.

Table 2 displays the estimated location errors at 17 testing locations with different downsampling rates α on the *n* reference images. It is difficult to determine whether this downsampling compromises the estimated location precision just reading this table. Thus, we utilize Equation (Equation 4)
(4)$$p=\frac{1}{17\times 9\times 4}\sum_{i=0}^{16}\sum_{j=0}^{8}\sum_{k=j+1}^{8}I_{A}\left(E_{ij},E_{ik}\right),$$

Here, $I_{A}\left(E_{ij},E_{ik}\right)=\left\{\begin{array}{c} 1,\;\;if\;E_{ij}\leq E_{ik}\\ 0,\;\;\;\;otherwise \end{array}\right.$, where $E_{ij},\;E_{ik}$ are cells in row *i*, column j,k. This function computes the probability, denoted by a variable *p*, that a cell in Table 2 is not greater than any cell to its right in the same row. It iterates through each row and compares each cell to those to its right, counting 1 if the value on the left is not greater than that on the right, i.e., smaller errors on the left and larger errors on the right. After this iteration and normalization of the counted number, we obtained the probability p=0.72, which is greater than 0.5, indicating that under the frame downsampling setting, the location estimation error when using a denser map (a value on the left) is indeed generally smaller than that when using a sparser map (a value on the right).

However, this frame downsampling setting is insufficient to prove our hypothesis because it has little effect on the direction variance of the reference image database, which is crucial for image retrieval. Consequently, we applied the second direction downsampling setting to see if direction downsampling also compromises the estimated location precision. Table 3 displays the estimated location errors at 17 testing locations with different downsampling rates β on the *m* perspective images of each equirectangular frame. Using a function similar to Equation (Equation 4), we obtained p=0.65, which is also greater than 0.5, indicating that the direction downsampling also reduces the accuracy of the location estimation.

### 5.2. Direction Results

These data so far partially support our hypothesis that the denser the reference image database is in our system, the more accurate our location estimation. However, we still need to determine whether frame or direction downsampling compromises the accuracy of the direction estimation. Table 4 shows the mean direction estimation errors for different frame/direction downsampling settings.

In contrast to the location estimation error, this table indicates that the overall direction estimation error is negligible. This is because that PnP algorithm leverages 2D–3D point correspondences between the query image and the sparse map, which are less dependent on the density of the reference image database and therefore more robust. However, the PnP algorithm fails on a few testing points when α or β becomes large (especially for β). This is because when reference images are too sparse, it is difficult for the system to find sufficient 2D–3D point correspondences, because few or no candidate images have overlap views with the query image.

## 6. Discussion

### 6.1. Technical Underpinnings of Navigation Solutions

Broadly speaking, navigation methods for assistive technologies can be categorized as sensor-based or vision-based. Localization for most sensor-based systems is highly accurate, but suffers from power consumption and deployability concerns. Although a handful of sensor-based navigation solutions have lower power consumption and are able to be deployed on mobile devices, most require pre-installed and carefully calibrated physical sensor infrastructure, which is costly, time-consuming, and often infeasible at-scale. To overcome these problems, our system employs a vision-based localization system that only requires commonly used cameras for data capture and can provide comparable accuracy on the location and direction estimation.

Nowadays, the great majority of sensor-based systems are difficult to install in large-scale outdoor environments, resulting in handoff concerns when navigation includes both indoor and outdoor environments. One of the conventional indoor localization technologies utilizes Radio Frequency Identification (RFID), which works with inexpensive tags and needs specific “antennas” to detect them. RFID is most commonly used for real-world position tracking, but when accuracy is required, it can be prohibitively expensive and implementation can be difficult and time-consuming. Bluetooth low energy (BLE) is one of the newest indoor localization technologies. Similar to RFID, it utilizes inexpensive tags, but is easier to install. It is not severely impacted by barriers. Nonetheless, its accuracy is typically quite poor (2–3 m). Ultra-wideband (UWB) is another recent indoor positioning technology, and is the most commonly used solution. It exploits a very low energy level for short-range, high-bandwidth communications throughout a large fraction of the radio spectrum. The UWB indoor navigation system is permitted without a license due to its low power. Compared to conventional signal systems such as RFID, UWB systems are more effective at penetrating obstacles. Due to its advantages, numerous UWB radio sensor-based indoor localization systems with centimeter-level precision have been adopted in the field [44]. However, UWB has a high risk of interfering with other systems that transmit in the ultra-wide spectrum due to configuration errors. Moreover, UWB receivers require extremely precise signal acquisition, synchronization, and tracking in relation to the pulse rate, which is time-consuming. Due to these shortcomings, only a tiny number of UWB integrated circuits for positioning systems are produced. All of these sensor-based indoor solutions are impractical to deploy them in large, complicated outdoor environments, where GPS has been widely used. GPS signal suffers from larger errors due to multi-path or ‘urban canyon’ [45], and it is challenging to obtain signal indoors. Compared to these sensor-based solutions, our system offers natural advantages in terms of power efficiency and immunity to electromagnetic interference, as it relies on commonly used cameras for data capture. It is easily deployable in both indoor and outdoor environments, and provides transitions between them, obviating the need to translate approaches from one sensor to another. Moreover, humans require a deployable solution over localization accuracy, so our approach is more ideal for supporting BLV navigation. Consequently, despite the fact that our system’s localization accuracy is slightly lower than the some of these sensor-based systems, it is still the preferred option for BLV navigation support.

Vision-based localization systems are not prone to electromagnetic interference, unlike sensor-based systems. One type of vision-based localization solution installs static cameras at specific locations throughout a building to track the BLV using artificial intelligence technologies [46,47,48]; these solutions can only be deployed in indoor environments and require precise calibration. Vision-based localization systems that employ a mobile camera and an image database [49,50] or a reconstructed 3D model [51,52] offer a robust path forward without the need for pre-installation, but these solutions require intrinsic camera parameters. Obtaining these parameters is logistically challenging, especially for those with BLV. Without such information, location estimation errors are frequently very large. Herein, we employed a novel approach with a weighted average algorithm to solve this challenge; it can begin working accurately with a sparse ma and, as the user continues to employ the system and thus increases the map’s density, the map evolves and the localization accuracy continuously improves.

### 6.2. Practical Implications

Our system is underpinned by video recordings that take on average 30/10,000 (min/sq feet), and generates a respective topometric map with registration between 2D and 3D in approximately 15 min, significantly less time than competing sensor-based solutions that require manual annotation. Moreover, our system can work jointly with janitorial (cleaning) robots and other citizen science opportunities to collect the relevant data required to generate the maps a priori. The boundaries/destinations GUI enables the map-maker to easily update information regarding map boundaries and destinations, allowing the system to rapidly adapt to changing environments, which is difficult for other vision-based and sensor-based approaches. In addition, our system can function without a cell signal by moving all computation onto the edge device; in other words, a relevant map of interest can be downloaded in advance, and the audio instructions can help people with BLV safely reach their destination of interest. Our system achieves positional and directional errors of 1 m and $2^{\circ }$, respectively, according to our evaluation, a considerable advance over other vision-based methods [25].

### 6.3. Limitations and Future Directions

We anticipate that our system will require a considerable amount of time to exhaustively search for similar reference images when databases grow to the size of a city; consequently, we could replace the existing method with a more advanced searching algorithm, such as KD-tree, to improve localization efficiency. In addition, even though the evaluation demonstrates that our system could achieve accurate localization with an average error of less than 1 m in a large indoor space if the database is sufficiently dense, we believe we can reduce the localization error even further if we can estimate the camera direction without intrinsic parameters. In [53], the author presents an implicit distortion model that enables optimization of the six-degree-of-freedom camera pose without explicitly knowing intrinsic parameters. In our future work, we will integrate this method, perform additional evaluations, and consider future pipeline upgrades. Finally, our system was evaluated in an indoor environment; however, its performance in an outdoor environment has not yet been determined. It is difficult to obtain an accurate floor plan for outdoor spaces. Reference [54] proposes an attention-based neural network for structured reconstruction for use in outdoor environments. The pipeline takes a 2D raster image as input and reconstructs a planar graph representing the underlying geometric structure; this new approach may afford us the ability to use satellite images to generate floor plans for use in our future work.

## 7. Conclusions

Herein, a prototype vision-based localization system has been introduced. This system does not require any pre-installed sensor infrastructure or a camera’s intrinsic matrix, making it superior to other *state-of-the-art* solutions for guiding the BLV in navigation. At present, our system uses a 360 camera to collect the initial reference image database in the mapping phase; then, simple cell phones are used to acquire additional image frames from multiple vantage points and create denser maps. The localization phase of the system only requires a daily-use camera, as found in most smart phones and tablets. The system is fashioned into a mobile application that can be downloaded on any smart device equipped with a camera and internet connection or onto ergonomic wearables, as illustrated by our novel backpack embodiment. Our goal for this approach is to support navigation of short and long length in both indoor and outdoor environments, with seamless handoffs. In the future, such a system could support additional microservices, such as obstacle avoidance or drop-off detection, evolving state-of-the-art wayfinding to a more integrated approach that blends orientation with travel support. 

## Figures and Tables

**Figure 1 sensors-22-08894-f001:**
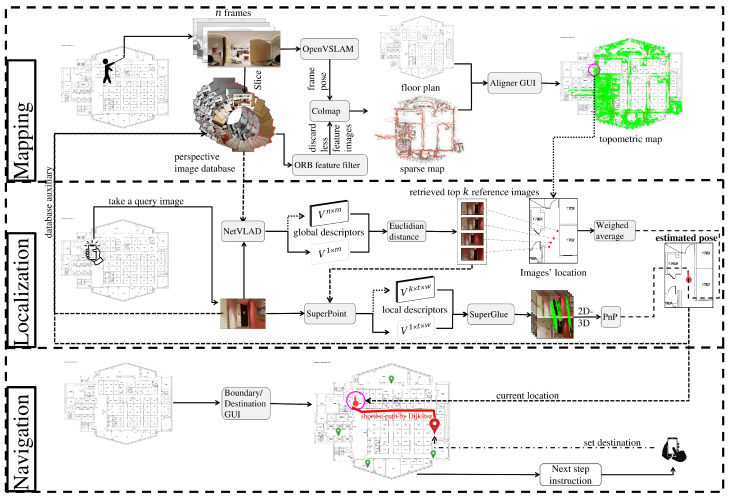
**System architecture diagram.** Our pipeline consists of mapping, localization, and navigation modules. During the mapping module, OpenVSLAM and Colmap algorithms use 360-degree images as input to generate a topometric map. On the basis of this topometric map, NetVLAD will first retrieve images similar to a query images; this is followed by two downstream tasks: weight averaging and coarse-to-fine PnP algorithms, which estimate the location and direction of the query image. In the navigation module, Dijkstra’s algorithm computes the shortest path between any database images using a navigable graph defined by the topometric map’s boundaries. During real-time navigation, our system will utilize both the shortest-path information and predefined destinations to guide end users to their desired destination. The captured query image will be returned to the mapping module to aid in the evolution of the topometric map.

**Figure 2 sensors-22-08894-f002:**
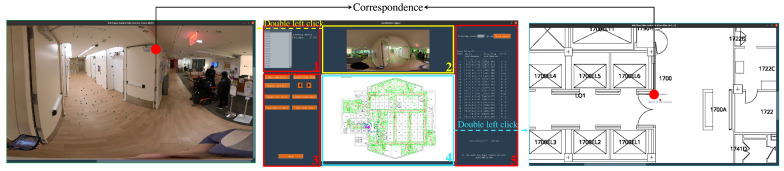
**Topometric map-aligner GUI.** The main window is displayed in the middle, which comprises five zones. *Zone 1* has a list of all equirectangular images recorded in the target space, which can be browsed in *Zone 2*. Several buttons in *Zone 3* aid the map-maker in projecting the raw map onto the floor plan. The map-maker can upload and display the floor plan of the target space in *Zone 4* by clicking the ‘Select Floor Plan’ button in *Zone 3*. Two magnified views (**left** and **right**) aid the cartographer in locating the 2D–3D correspondences between the equirectangular image and the floor plan. Once the transformation matrix is identified, *Zone 5* will display map error information.

**Figure 3 sensors-22-08894-f003:**
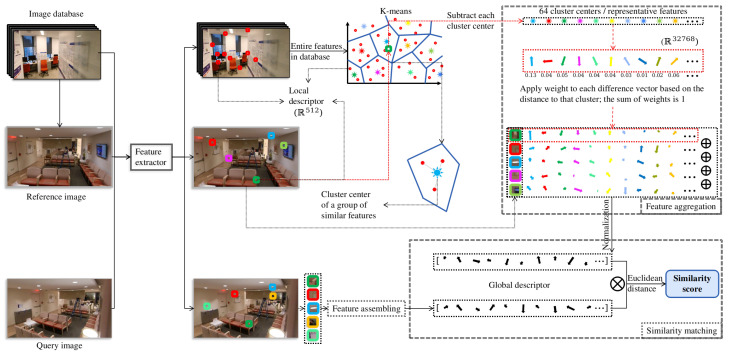
**Retrieval-based localization.** This diagram demonstrates the steps of feature aggregation and similarity matching using NetVLAD. A pretrained NetVLAD extracts several local descriptors of the query image and each reference image, assembling them into global descriptors and calculating the Euclidean distance between the query image’s global descriptor and that of each reference image; the smaller the Euclidean distance is, the more similar the images are.

**Figure 4 sensors-22-08894-f004:**
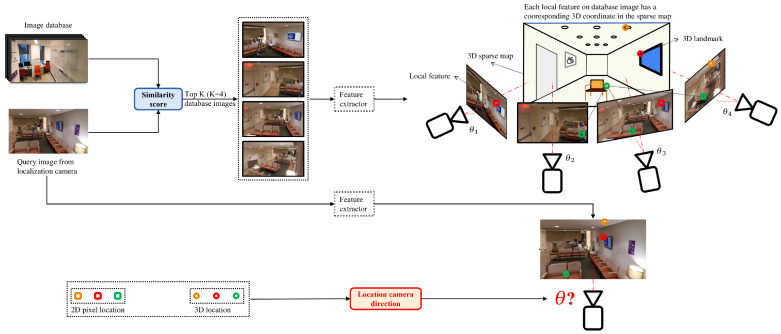
**Hierarchical localization**. Given a query image, the K reference images with the highest matching scores (Figure 3) are retrieved as candidate images. Using SuperPoint and SuperGlue, similar local features between the query image and candidate images can be identified. Each local feature on the candidate image has a 3D location in the sparse map. Thus, a set of 2D–3D point correspondences between the query image and the sparse map are found and the direction of the camera is determined using the PnP algorithm.

**Figure 5 sensors-22-08894-f005:**
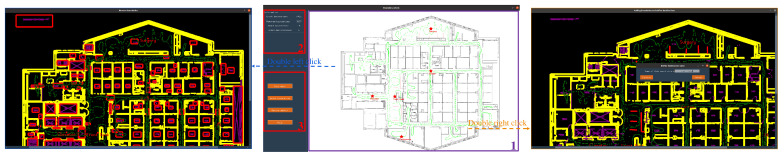
**Boundary/destination GUI.** The main window is displayed in the **middle**. It comprises three zones. The green dots in *Zone 1* depict the locations of database images, while the red star indicates potential destinations. After double-clicking *Zone 1*, map-makers can adjust boundaries and desired destinations in additional pop-up windows (**left** and **right**). After double left-clicking *Zone 1*, the map-maker can delete boundaries in the pop-up window (**left**) by removing line segments (which represent boundaries) using an edit tool (red superimposed box). After double right-clicking *Zone 1*, another pop-up window (**right**) will appear, in which the map-maker can draw red lines to represent additional boundaries and define potential destinations by clicking green dots and assigning destination names to them. Additionally, boundaries and destination information are provided in the *Zone 2* (**middle**). Once all the boundaries and destinations are defined, the map can be saved in *Zone 3*.

**Figure 6 sensors-22-08894-f006:**
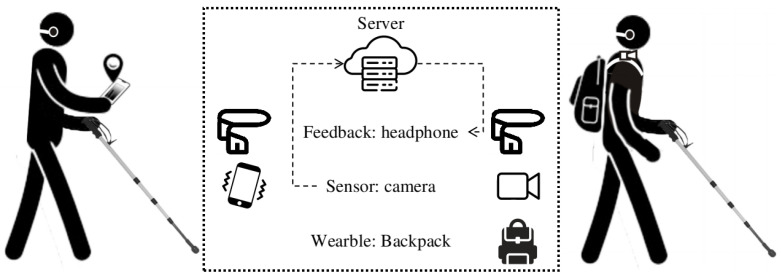
**Two types of end-user interfaces.** On the left is an infographic that demonstrates the use of our Android application. The user selects the desired destination and sends it to the server. For localization, the end user touches the screen (or waits for the time interval) to capture a query image, which is sent to the cloud server. The server then calculates the camera pose and sends instructions to the end user via a binaural bone-conduction headset. On the right is an infographic that demonstrates the use of our wearable device with a binaural bone-conduction headset. During navigation, a camera connected to a backpack-mounted NVIDIA${}^{®}$ Jetson AGX Xavier captures and sends query images to a cloud server; the user receives navigation instructions via the headset.

**Figure 7 sensors-22-08894-f007:**
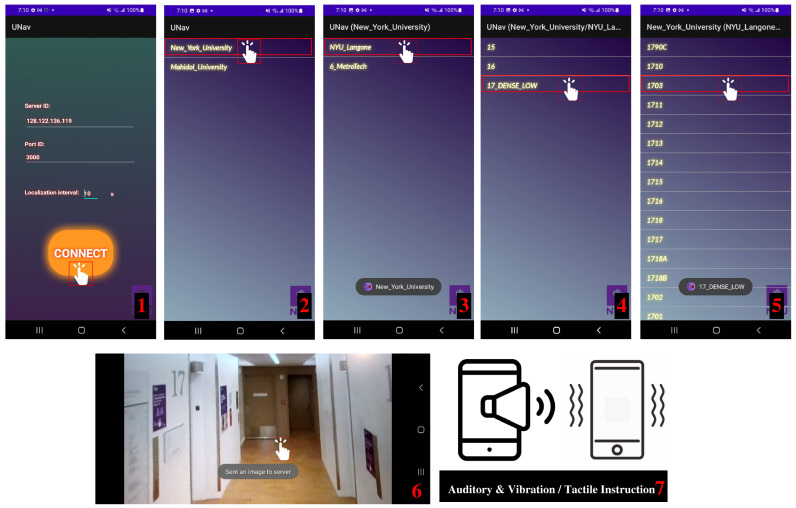
**Android application.** There are seven steps to any trip initiation. The first step (1) is to enter the server ID, port ID, and the time interval between two image captures/server instructions (the end user can use the default settings without any changes). The next four steps (2–5) are a cascade of linked choices that must be completed for the user to select the desired destination, inclusive of place, building, floor, and room. This procedure may be completed with either verbal or tactile input. Once the destination has been selected, the camera will turn on and the user will need to wait for the capturing time interval of touch the screen to take and send an initial query image to the server (Step 6). The server will then send back navigation instructions via auditory or vibration signals (Step 7). After multiple iterations of Steps 6 and 7, the user will reach the desired destination.

**Figure 8 sensors-22-08894-f008:**
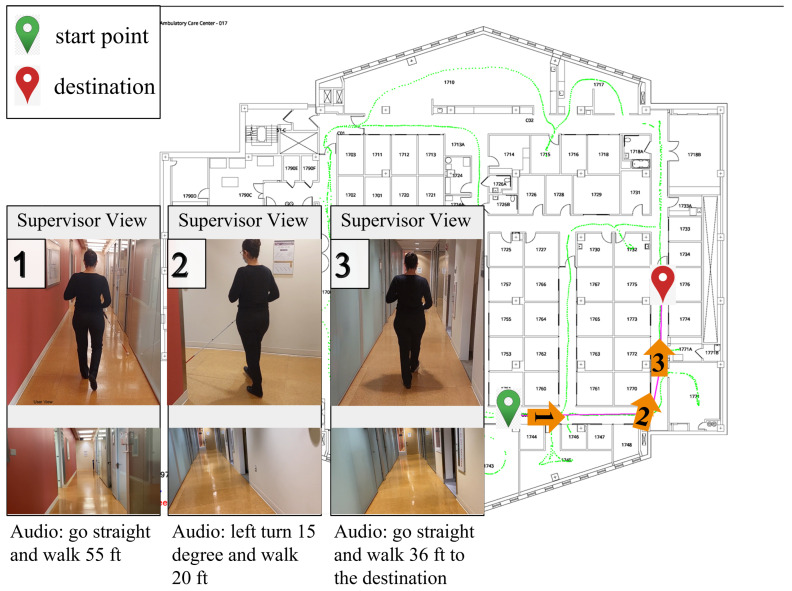
**Navigation example.** Using our Android application, a member of our team navigates from the origin to the destination safely.

**Figure 9 sensors-22-08894-f009:**
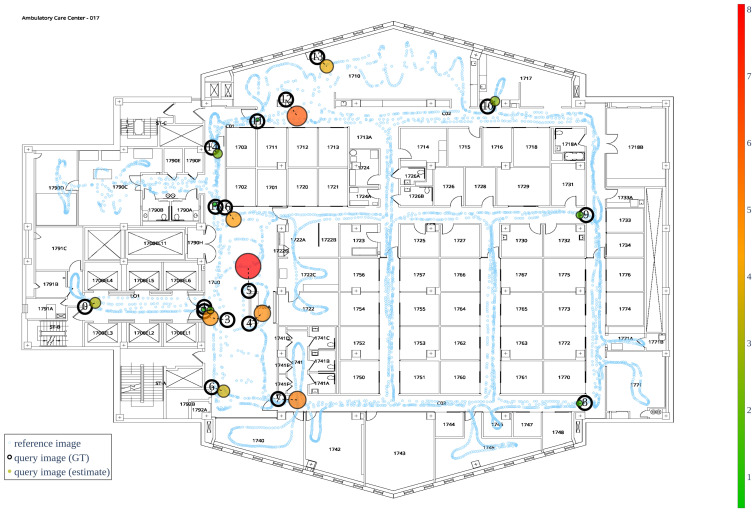
**Location accuracy heatmap.** The small open light-blue dots are the locations of *n* reference images on the floor plan. The open black circles are the ground truth locations of 17 testing images; the multi-colored circles next to the GT localizations colored from green to red are the estimated locations of these 17 testing images; the further from the ground truth location (ft), the larger the circle and the warmer the color (red).

**Table 1 sensors-22-08894-t001:** **Advantages of our methods compared with OpenVSLAM and Colmap.** Our approaches use sequential image inputs and support both the equirectangular camera model and the SuperPoint feature, resulting in strong mapping and localization performance.

	OpenVSLAM	Colmap	Our Methods
Sequential image inputs(Robust in mapping)	✔		✔
Support equirectangular camera model(Robust in mapping)	✔		✔
Support SuperPoint feature(Robust in localization)		✔	✔

**Table 2 sensors-22-08894-t002:** Estimated location errors (ft) at *17* testing locations with different frame downsampling rates $\alpha \in \left[1,5,10,15,20,25,30,40,50\right]$ on n=4258 reference images.

Different Frame Sampling Density										
	**Error (ft)**	n/1	n/5	n/10	n/15	n/20	n/25	n/30	n/40	n/50
Different Testing Locations	0	3.1	3.1	2.5	2.4	2.6	2.6	2.7	3.1	3.8
	1	0.9	1.6	1.6	1.7	1.5	2.4	1.1	1.4	1.4
	2	1.8	2.3	2.1	3.7	3.2	9.2	3.4	2.0	7.3
	3	4.8	4.2	3.1	4.9	1.2	5.1	4.6	5.1	5.7
	4	5.0	3.3	4.3	5.9	3.4	7.2	5.9	11.1	7.1
	5	8.1	9.1	10.7	5.3	10.4	8.1	10.7	4.9	8.6
	6	3.6	4.3	4.5	4.1	5.3	4.2	14.4	4.7	20.2
	7	5.4	5.4	5.4	5.4	11.8	53.0	6.0	8.5	6.5
	8	1.5	3.0	1.8	5.5	17.4	2.7	17.3	17.1	14.2
	9	1.8	3.3	3.3	1.6	24.6	4.1	22.9	22.3	18.9
	10	2.7	1.9	3.0	3.0	3.2	3.0	3.0	3.2	3.0
	11	0.5	0.7	0.7	4.6	0.7	4.8	53.2	0.7	11.6
	12	6.2	3.7	3.7	8.3	8.6	10.2	8.3	8.6	10.2
	13	4.0	17.7	5.4	8.4	5.8	5.0	4.6	3.4	8.0
	14	2.5	2.3	3.1	3.7	2.8	1.9	3.7	12.7	4.5
	15	0.9	1.2	2.5	10.1	2.5	14.2	45.6	2.5	12.1
	16	4.6	3.7	5.4	5.1	6.2	17.0	5.8	4.8	20.1

**Table 3 sensors-22-08894-t003:** Estimated location errors (ft) at *17* testing locations with different direction downsampling rates β∈[1,2,⋯,6] on $\hat{m}\leq 18$ filtered perspective images of each equirectangular image.

Different Frame Sampling Density							
	**Error (ft)**	m/1	m/2	m/3	m/4	m/5	m/6
Different Testing Locations	0	3.1	2.8	2.8	3.0	4.1	4.5
	1	0.9	0.9	1.8	3.0	3.8	3.8
	2	1.8	3.4	3.4	3.8	7.1	2.7
	3	4.8	5.3	4.0	2.6	4.2	5.2
	4	5.0	5.4	5.0	7.5	1.3	8.0
	5	8.1	7.9	8.1	9.3	8.1	9.1
	6	3.6	4.1	5.3	4.0	7.9	1.5
	7	5.4	2.1	5.4	5.4	5.4	112.3
	8	1.5	2.1	1.8	1.7	1.4	1.7
	9	1.8	2.1	1.8	2.1	1.1	1.1
	10	2.7	2.7	2.1	2.7	2.1	2.1
	11	0.5	0.5	0.7	3.0	0.6	39.6
	12	6.2	4.3	4.6	0.8	7.3	5.8
	13	4.0	3.9	3.7	17.7	17.7	15.9
	14	2.5	3.1	2.9	2.7	2.5	1.8
	15	0.9	0.9	0.9	0.9	0.9	0.9
	16	4.6	5.8	5.4	5.1	5.7	5.2

**Table 4 sensors-22-08894-t004:** Mean estimated direction errors (${}^{\circ }$) of 17 testing locations under different combination of frame downsampling rate α and direction downsampling rate β. n/s indicates that directions at some testing locations cannot be retrieved.

Different Direction Sampling Density							
	**Error (ft)**	m/1	m/2	m/3	m/4	m/5	m/6
Different Frame Sampling Density	n/1	2	3	3	2	2	13
	n/5	2	2	2	2	3	3
	n/10	2	2	2	2	2	2
	n/15	3	3	n/s	3	3	n/s
	n/20	2	3	2	3	3	3
	n/25	4	3	3	4	4	14
	n/30	3	3	n/s	3	3	n/s
	n/40	2	3	2	3	2	n/s
	n/50	4	3	n/s	n/s	4	n/s

## Data Availability

The data presented in this study are available in our google drive (https://drive.google.com/drive/folders/1xhGmdxgGzY0HCikQWW7MyAA2vF-MWyux?usp=sharing). accessed on 17 August 2022.

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
