# Peer review of "UNav: An Infrastructure-Independent Vision-Based Navigation System for People with Blindness and Low Vision"

_sensors, 2022, doi:10.3390/s22228894_

Round 1
Reviewer 1 Report
Suggestions:
1. Justify the reason for using Dijkstra's algorithm and not another algorithm.
2. To test the system’s overall localization accuracy, explain in more detail why 17 points were specifically selected on the floor plan as testing locations.
Reviewer 2 Report
This is a good work on the application of existing algorithms for indoor topogeometric navigation. However, some points are missing, such as detailing the time required to take the images for the mapping, or in the execution of the algorithms. It would have helped to have a greater number of people in the study and to know their opinion about it. Finally, there is a lack of an error detection mechanism for false positives that would initiate erroneous navigation.
Reviewer 3 Report
The development of assistive technologies presents numerous challenges, which have been overcome within the various research developed in the area, some starting with modeling, and space signaling (https://www.mdpi.com/1424-8220/20/9/2641)( https://www.mdpi.com/1424-8220/13/1/241/htm), and others by the development of assistive instruments that help in the navigation of PD in indoor environments (https://www.mdpi.com/2411-9660/5/4/75).
In section I (Introduction), a greater contextualization of the research scenario is needed. Already in section II. (Related Works) it would be interesting to insert a table with research data/parameters, and current systems that can help to identify the contribution achieved by the authors in relation to the solutions/research that already exist.
page 4, improve the resolution of figure 1.
page 5, Table 1 is little
page 10, figure 6 needs improvement, the illustration is not very representative.
page Figure 7 (1, 2, and 3) can be improved...
Sections 6 and 7 need to be improved, a more objective discussion needs to be addressed in relation to the most used methods for locating people and objects in indoor environments.
Round 2
Reviewer 3 Report
In general, the observations about the manuscript were met, I would only ask to improve the resolution of figure 9.
I don't think I have any further observations...